# Fabrication of Robust Superhydrophobic Surfaces with Dual-Curing Siloxane Resin and Controlled Dispersion of Nanoparticles

**DOI:** 10.3390/polym12061420

**Published:** 2020-06-25

**Authors:** Hyeran Kim, Kibeom Nam, Dong Yun Lee

**Affiliations:** Department of Polymer Science and Engineering, Kyungpook National University, Daegu 41566, Korea; rwnka110@naver.com (H.K.); ska2918@naver.com (K.N.)

**Keywords:** superhydrophobic coating, spray coating, robustness and stability, nanoparticles

## Abstract

We developed a simple method for the fabrication of superhydrophobic surfaces on various substrates using spray coating. The fabrication method started with the blending of a modified hydrophobic siloxane binder, silica nanoparticles, and a volatile solvent by sonication. The mixture was spray-coated on various surfaces such as slide glass, paper, metal and fabric, forming a rough surface comprising silica particles dispersed in a hydrophobic binder. Surface hydrophobicity was affected by the surface energy of the binder and the degree of roughness. Therefore, we realized a superhydrophobic surface by controlling these two factors. The hydrophobicity of the siloxane binder was determined by the treatment of fluorine silane; the roughness was controlled by the amount of coated materials and sonication time. Thus, using the spray coating method, we obtained a superhydrophobic surface that was mechanically durable, thermally stable, and chemically resistant.

## 1. Introduction

Surface wettability control has attracted considerable attention over the past decades. Superhydrophobic surfaces (e.g., surfaces of lotus leaves) are frequently found in nature. They are of particular interest because their properties are useful for self-cleaning [1,2,3] anti-icing [4,5], liquid separation [6], and anti-corrosive [7] applications. A superhydrophobic surface is a surface with a water contact angle (CA) greater than 150° and a water sliding angle (SA) below 10°. Superhydrophobicity arises from the combination of low surface energy and hierarchical structure with dual or multiscale surface roughness [8,9,10,11]. Superhydrophobicity has been explained by the Wenzel and Cassie–Baxter theoretical models [12,13]. On a rough hydrophobic surface in the Wenzel state, grooves on the surface are completely filled with water. In the Cassie–Baxter state, air pockets are present between the surface and water droplets. The wetting properties between water droplets and the substrate considerably differ between the two states. In the Wenzel state, water droplets are pinned to the surface and do not roll off easily. Droplets are likely to fall from the surface in the Cassie–Baxter state because they sit partially on air.

Several researchers have endeavored to develop fabrication techniques for superhydrophobic surfaces with multiscale roughness, including phase separation [14], plasma or chemical etching [15,16], casting [17], self-assembly [18], and lithography [19,20,21]. These methods often involve multiple complex processing steps, costly reagents, and expensive equipment. Limitations in terms of substrate size and materials used may also exist. However, spray coating is a simple and inexpensive process that is not limited by the substrate. Therefore, superhydrophobic surfaces with multiscale roughness are frequently fabricated by spraying nanoparticles (NPs) and a binder onto the substrate. The fabrication of superhydrophobic surfaces by spray coating involves the consideration of various factors, such as the size and composition of NPs, binder content, and amount of coating solution because they affect the wettability of the surface [22,23]. Controlling the size of NP aggregates in solution is particularly important because they affect the surface morphology of the resultant coating. Particle aggregation can be controlled by selecting an appropriate solvent such as ethanol, propanol, or butanol [24]. However, few studies have addressed how NP dispersion affects the wetting properties of surfaces related to NP aggregation.

For practical use, a superhydrophobic coating must be mechanically durable. Microscale and nanoscale structures on a superhydrophobic surface can be easily destroyed and removed by mechanical abrasion, water impact, or even with the touch of a finger. Such damage to a superhydrophobic surface results in the loss of its non-wetting properties. It is difficult to fabricate a durable superhydrophobic surface by spray coating because there are many interfaces between the coating and substrate. To address this issue, Wu et al. [25] fabricated a repairable, fluorine-free superhydrophobic surface. The surface could be simply regenerated by repeating the spray deposition process. Sparks et al. [26] obtained superhydrophobic inorganic–organic thiol-ene coatings by sequential spray deposition and photopolymerization. The wettability of the coatings was unaffected by normal handling owing to the crosslinked thiol-ene polymer network. However, wettability was tested only by pressing the surfaces with a gloved finger. Lee et al. [22] reported the fabrication of robust superhydrophobic surfaces by depositing a polydimethylsiloxane (PDMS) layer onto spray-coated substrates. Zhang et al. [27] enhanced the mechanical robustness of superhydrophobic coatings by spraying silica NPs on a chemically etched aluminum alloy. However, the superhydrophobic surfaces were not robust and the fabrication process was complex.

Herein we describe a simpler and more efficient method for the fabrication of mechanically durable superhydrophobic surfaces. We prepared a robust siloxane resin as a spray-coating binder and controlled surface morphology with NPs. This method does not require a hydrophobic modification for NPs dispersion and a post-treatment process to lower the surface energy of the coated film. The dual-curing resin was synthesized via the hydrolysis and condensation of silanes and cured with amino silanes to increase durability. The surface morphology of spray-coated samples was regulated by controlling the dispersion of NPs in the coating solution by sonication time and the concentrations of coating materials. The fabricated superhydrophobic surfaces were exceptionally water repellent and highly robust. We performed pencil hardness and sand abrasion tests to evaluate the mechanical properties of the coatings. The superhydrophobicity of the coatings was maintained even at 300 °C, and they were highly resistant to various organic solvents.

## 2. Materials and Methods

### 2.1. Materials

(Heptadecafluoro-1,1,2,2-tetrahydrodecyl)triethoxysilane (FAS) was obtained from Gelest (Morrisville, PA, USA). (3-Glycidyloxypropyl)trimethoxysilane (GOTMS), (3-aminopropyl)triethoxysilane (APTES), an ammonium hydroxide solution, and 15–30 nm (ø) fumed silica NPs were purchased from Sigma-Aldrich (St. Louis, MO, USA). Absolute ethanol was purchased from Duksan Inc. (Ansan, South Korea).

### 2.2. Preparation of Dual-curing Siloxane Resins with Low Surface Energies

Modified siloxane resins were synthesized using a base-catalyzed hydrolytic condensation reaction [28,29,30,31,32]. The mixtures of GOTMS and 0, 1.0, 1.3, 2.0, 4.0, or 8.0 mol % FAS were prepared and stirred for 5 min. Water and an ammonium hydroxide solution were added to the mixtures under constant stirring. Water was used to accelerate hydrolysis, and ammonium hydroxide was used as a catalyst to initiate the sol–gel process. GOTMS and FAS hydrolysis and condensation were performed at room temperature for 24 h to form siloxane bonds. Siloxane resins containing fluoroalkyl groups were obtained as white suspensions and used as hydrophobic binders. Different amounts of APTES were added to the resulting siloxane resins to obtain GOTMS:APTES molar ratios of 1:0.1, 1:0.25, 1:0.5, 1:1, and 1:2 when APTES was poured, and the resins were cured at 150 °C for 2 h.

### 2.3. Fabrication of Superhydrophobic Surfaces by Spray Coating

Each spray deposition mixture was prepared by dispersing a modified siloxane resin and APTES in ethanol (10 g) and adding NPs in different ratios. The suspensions were stirred for 5 min and sonicated before use. Each substrate (76 mm × 26 mm) was coated with 2 mL of a colloidal silica suspension using an airbrush (Iwata Eclipse, ANEST IWATA Corporation, Yokohama, Japan) with a nozzle diameter of 0.35 mm. The airbrush was operated with compressed nitrogen gas (1 bar). The distance between the airbrush tip and the substrate was approximately 15 cm. The silica NP spray-deposition process and annealing of a spray-coated sample are illustrated in Figure 1a. When droplets sprayed from the nozzle reached the substrate, the NPs tended to aggregate owing to rapid alcohol evaporation. The randomly distributed aggregates on the substrate formed microscale and nanoscale bumps. The chemical properties of the ethanol did not affect the hydrophobicity of the coatings because it evaporated when the sample was spray-coated and annealed.

### 2.4. Characterization

The diameters of the particles were measured by dynamic light scattering (DLS) analysis using a Scatteroscope I particle size analyzer (K-One Nano, Seoul, South Korea). The elemental compositions of modified siloxane resins and the resulting superhydrophobic surfaces were determined by X-ray photoelectron spectroscopy (XPS) on a PHI Quantera SXM XPS instrument (Ulvac-Phi, Osaka, Japan). Gel permeation chromatography (GPC) was performed on an Alliance e2695 Separations Module (Waters, Milford, MA, USA) at 35 °C with polystyrene standards to determine the molecular weights of the siloxane resins. The retention time was 30 min, and tetrahydrofuran (THF) was used for elution. The CAs of 5-μL water droplets on the spray-coated samples were measured with a GSX contact angle meter (Surfacetech, Kansas City, MO, USA). SAs were measured using the tilting stage method with 15-μL water droplets. All measurements were repeated at least five times in different sampling locations. The surface morphologies of the coatings were inspected using an SU-8220 scanning electron microscope (SEM, Hitachi, Tokyo, Japan) at an electron acceleration voltage of 5 kV. To analyze the surface roughness of the spray-coated samples, the root-mean-square roughness value (Rq) was determined using an LSM-700 confocal laser scanning microscope (Carl Zeiss, Oberkochen, Germany). Thermal stability was evaluated using a Q500 thermogravimetric analyzer (TGA, TA Instruments, New Castle, DE, USA). Tests were performed at a heating rate of 10 °C/min from 30 to 700 °C. The mechanical durability of as-prepared coatings was investigated by conducting sand abrasion tests using commercial sand with particle diameters ranging from 100 to 300 μm. The samples were held at 45° relative to the horizontal surface, with the sand impacting their surfaces at a rate of 10 g/min from a height of 30 cm for 30 min. The samples’ mechanical durability was also evaluated in pencil hardness tests performed according to ASTM D3363. Pencils ranging from 6B to 9H on the hardness scale were purchased from Mitsubishi (Tokyo, Japan) and applied with a loading of 500 g at 45°.

## 3. Results and Discussion

### 3.1. Dual-curing Siloxane Resins with Low Surface Energies

The synthesis and curing of fluorinated siloxane resins are illustrated in Figure 1b. Siloxane bonds were formed in the resins during the base-catalyzed hydrolysis and condensation of GOTMS and FAS. The obtained siloxane resins were dual-cured, which involved reactions between alkoxy groups in the resins and APTES and between epoxy groups in the resins and reactive hydrogen atoms in APTES. GOTMS:APTES molar ratios of 1:0.1, 1:0.25, 1:0.5, 1:1, and 1:2 were evaluated to systematically investigate the effect of variation of the APTES content. Reference samples were prepared by spin-coating glass substrates at 2000 rpm for 30 s. Resins with GOTMS:APTES ratios of 1:0.1 and 1:0.25 contained small amounts of APTES. The films obtained with these resins did not form well, and the glass substrates were exposed (Appendix A). This was presumably owing to the small amounts of APTES; the low surface energies of the resins enabled rapid aggregation before crosslinking. The 1:0.25 GOTMS:APTES sample was more uniformly coated than the 1:0.1 GOTMS:APTES sample. With greater amounts of APTES at GOTMS:APTES ratios of 1:0.5, 1:1, and 1:2, the resins were uniformly coated (Appendix A). The surface hardness of each coated sample exceeded 9H, and the samples had nearly equal CAs of ~105°.

The GOTMS:APTES molar ratio of 1:0.5 was determined to be optimal; thus, the corresponding resin was selected for subsequent experiments. XPS analysis showed that fluorine was successfully incorporated into the siloxane resin (Appendix A). The XPS spectrum of the sample containing 2 mol % FAS was similar to that of the 0 mol% FAS sample, and Si 2s and Si 2p peaks were observed in each sample. An additional F 1s peak appeared in the spectrum of the 2 mol% FAS sample, which confirmed that its surface contained fluorine. The non-wetting properties of fluorinated resins arose from their low surface energies, which were because of the presence of fluorine atoms in the resins. A significant amount of fluorine was detected on the upper surface of the 2 mol % FAS sample (Appendix A).

The surface energies of fluorinated siloxane resins can be controlled by adjusting the FAS concentration. The surface energies of resins with varying FAS content were determined by the Fowkes method using diiodomethane and deionized (DI) water. The surface energies are plotted in Figure 1c as a function of FAS content and are summarized in Appendix A. As expected, the surface energies of the resins decreased with increasing FAS content and reached a saturation value of ~19.8 mJ/m². Compounds that contain fluorine are generally expensive and are not very soluble in common organic solvents [33]. Considering cost and solubility, 2 mol % FAS was determined as the optimal concentration for resin synthesis. The molecular weight (MW) and MW distribution of the siloxane resin prepared with 2 mol % FAS 2 were determined via GPC. The weight average MW and the polydispersity index of the sample were ~3000 g/mol and 1.35, respectively. A pencil hardness test was performed to examine the mechanical durability of the hydrophobic resin. The thickness of the cured resin on the glass substrate was ~50 µm, and it remained intact after testing with a 9H pencil. Thus, the fluoroalkyl-containing siloxane resin is a suitable binder for the fabrication of mechanically durable superhydrophobic surfaces owing to its low surface energy and excellent durability.

### 3.2. Wetting Behavior

A solid surface’s wettability of generally depends on its chemical composition and surface morphology. The CA of any flat surface cannot surpass 120° even if its surface energy is low [34,35]. Thus, surface roughness is essential for a non-wetting surface. A surface can be easily roughened by spray coating it with a colloidal silica solution. Several spray-coating variables govern the hydrophobicity of coatings. It was determined that the aggregation of silica NPs in the coating solution considerably affects the wetting behavior of a spray-coated surface. As expected, the size of NP aggregates in our solutions depended on the sonication time. The aggregates decreased in size with an increase in the sonication time (Figure 2a and Appendix A). When the sonication time increased from 1 to 90 min, the average size of the silica NP aggregates decreased from ~500 to ~240 nm.

The wetting properties of the spray-coated surfaces were investigated after sonication for different amounts of time by measuring the CAs and SAs of water droplets. The concentrations of the silica NPs and the binder were fixed at 1 and 5 wt. %, respectively. Upon increasing the sonication time CAs decreased and SAs increased (Figure 2b). This indicated that long sonication times reduced the hydrophobicity of coated surfaces, and the coated samples were not superhydrophobic when the sonication time exceeded 10 min. Surface wettability depends on surface energy and roughness. In this study, surface energy was fixed; thus, wetting behavior was governed by surface roughness.

The surface morphologies of coatings fabricated after sonication for various times were investigated by SEM and confocal laser scanning microscopy (Figure 2c–h). The surfaces of the coatings became flatter with an increase in sonication time. The *R*q values after sonication for 1, 5, 10, 30, 60, and 90 min were 9.6, 4.4, 3.6, 1.9, 1.7, and 1.6 μm, respectively. These results reveal that superhydrophobicity was obtained with short sonication times. This occurred because shorter sonication times resulted in the formation of rougher hierarchical surfaces. We attributed the morphological differences to variations among silica NP aggregates in the coating solutions. As shown in Figure 2a, silica NPs became more dispersed with an increase in sonication time. The spraying of smaller NP aggregates likely generated relatively smooth surfaces. These results indicate that the wetting properties of spray-coated surfaces can be systematically controlled by varying the sonication time. Thus, it appears that shorter sonication times are conducive to the formation of multiscale roughness on the substrates. However, short sonication times result in superhydrophobic surfaces unsuitable as durable coatings. This occurs because the siloxane resin, which acts as a binder, cannot flow into the gaps between the particles. Microscopic bumps tend to be more easily damaged than smaller bumps. In addition, NPs in solutions sonicated for less than 30 s were insufficiently dispersed to pass through the narrow nozzle and clogged the tip. The sample sonicated for 1 min had a B–HB pencil hardness, while the sample sonicated for 5 min had a pencil hardness of H–2H (Appendix A). Therefore, the optimal sonication time to achieve both superhydrophobicity and durability was 5 min.

In addition to the sonication time, the NP content affected the hydrophobicity of the coatings. Figure 3a shows the CAs and SAs of water droplets on coatings with different silica NP contents. An increase in the NP content commonly results in a more hydrophobic spray-coated surface. After sonication for 5 min, samples with a binder content of 5 wt. % were most superhydrophobic when the NP content exceeded 1 wt. %. When the coating contained no NPs (0 wt. %), CA was approximately 105°. SA could not be measured because the hydrophobic siloxane resin had a nearly flat surface morphology. The SA of the sample coated with a 0.7 wt. % silica mixture was sharply decreased. When the particle concentration exceeded 1 wt. %, the droplets were in the Cassie–Baxter state. SA decreased by nearly 1°, and CA exceeded 150°. Thus, superhydrophobic surfaces had NP contents of 1 wt. % or higher. This occurred because surface roughness increased as the NP concentration increased (Appendix A). The *R*q values of the coatings were 2.6 (0.5 wt. % NPs), 3.2 (0.7 wt. % NPs), 4.4 (1.0 wt% NPs), 8.6 (1.5 wt. % NPs), 13.6 (2.0 wt. % NPs), and 33.4 μm (4.0 wt. % NPs). The coating was easily removed by even a slight impact if the particle content was ≥2 wt. % because the amount of binder between NPs was insufficient. Moreover, CAs on the surfaces were lower. Accordingly, it was assumed that the relatively hydrophilic silica NPs had greater contact with the surface than the hydrophobic binder.

XPS analysis revealed a steady decrease in the F/O ratios on the coated substrate from 0.491 to 0.030 when the particle concentration increased from 0.5 wt. % to 4.0 wt. % (Appendix A). Another variable affecting CA and SA was the amount of hydrophobic binder, which acted as an adhesive. The sonication time and silica NP content were held constant at 5 min and 1 wt. %, respectively. The surface coating prepared without the binder was completely wetted with water droplets because the large number of hydroxyl groups on the surface of silica NPs made them hydrophilic. The CAs and SAs of spray-coated samples with binder concentrations of 1, 3, 5, 7, and 10 wt. % are shown in Figure 3b. The trend observed with an increase in binder content was similar to that observed with an increase in NP content. Coatings with low binder concentrations had considerably more hydrophobic surfaces; however, the coatings were easily damaged. The pencil hardness of the coating prepared with 3 wt. % of the binder was 2B–3B (Appendix A). Silica NPs were entirely covered with the binder when the binder concentration was too high, and the resulting surfaces were relatively smooth. The hydrophobicity of the surfaces was lower owing to low surface roughness. On the basis of these results, the optimal fabrication conditions for mechanically durable superhydrophobic coatings include a sonication time of 5 min, silica NP concentration of 1 wt. %, and binder concentration of 5 wt. %.

### 3.3. Mechanical Durability

To investigate the resistance of superhydrophobic coatings to mechanical force, sand abrasion and pencil hardness tests were performed. Figure 4a shows the schematic illustration of a sand abrasion test. Sand was released and allowed to fall freely from a height of 30 cm onto substrates coated with samples prepared under optimal conditions (Section 3.2). The CA of water exceeded 150°, and SA fell below 10° after 30 min. However, the hydrophobicity of samples decreased only marginally. Sand abrasion results indicated that the fabricated superhydrophobic coatings were highly durable. Pencil hardness tests were performed to characterize the mechanical properties of spray-coated samples. The pencil hardness of spray-coated surfaces obtained under different conditions is summarized in Appendix A. Surface hardness decreased with decrease in the sonication time and binder content and increase in particle content.

Many applications require exposure of coated surfaces to heat. Thus, it is necessary to assess the thermal stability of superhydrophobic surfaces at high temperatures [36]. Therefore, the as-prepared samples were annealed for 1 h as the temperature increased from 25 to 400 °C. The CAs and SAs of the annealed samples are shown in Figure 4b. No changes in water CA or SA were observed at temperatures below 300 °C, indicating that the surfaces remained intact. CA considerably decreased, and superhydrophobicity was absent after annealing at 400 °C, which was due to the decomposition of the hydrophobic siloxane resin (Appendix A).

The resistance of the superhydrophobic coatings to organic solvents was assessed by exposing the samples to various environmental conditions. First, the spray-coated samples were submerged in common organic solvents (acetone, ethanol, isopropyl alcohol (IPA), THF, and toluene) for 200 h. The samples were completely dried in an oven at 60 °C before measuring CAs and SAs on the coatings. Interestingly, the superhydrophobic surfaces retained their non-wetting properties after immersion in all aforementioned solvents (Figure 4c). After 200 h of immersion, the surfaces of the samples showed no change. They appeared as dry as they were before immersion, and even tiny spots were not observed. Furthermore, to prove the stable non-wettability, CAs changes were measured while a water droplet remained on the superhydrophobic surface for 30 minutes (Appendix A). These samples exhibited excellent stability upon heating to temperatures up to ~300 °C. The resistance of as-prepared coatings to common organic solvents can be attributed to the robust siloxane matrix. Siloxane and C–F bonds are resistant to heat and chemicals because these bonds are strong. Bond strength is a crucial consideration in the design and fabrication of stable superhydrophobic coatings. Furthermore, durability is especially important under environmental conditions involving mechanical force, high temperatures, and/or corrosive substances. Durability can considerably extend the range of applications for a coating, such as water collection in a desert without loss of hydrophobicity.

### 3.4. Applications of Superhydrophobic Coatings

The hydrophobic coated surface in Figure 5a was prepared with the siloxane binder but without NPs. It was tilted at an angle of 15°, and the transport of water droplets over the surface was monitored. Water droplets on this coating strongly adhered to the surface and did not readily roll across it. When the water droplets gathered and gained mass, they tumbled over the surface owing to the force of gravity [37,38]. However, water droplets weakly adhered to the superhydrophobic coating prepared with silica NPs (Figure 5b). Water droplets on this surface were nearly spherical and rolled off rapidly. However, the sample was unsuitable for water collection. The movement of water droplets could not be controlled, and the droplets moved in all directions (Appendix A). An illustration of the superhydrophobic coating with hydrophilic guiding tracks (2 mm in diameter) is shown in Figure 5c. The hydrophilic pattern on the superhydrophobic coating was formed by scraping the coating with a sharp object. The pattern on the superhydrophobic coating transferred the water droplets from a relatively non-adherent rolling state to a strongly adherent pinned state. The water droplets gathered and combined. When the combined droplets reached a certain mass (10 μL), they flowed along the hydrophilic guiding tracks (Figure 5d and Appendix A). Thus, the hydrophilic patterns enabled us to collect water without loss. The sprayed coating could be etched with a hard, sharp point to expose hydrophilic areas to create a patterned surface. The sharp point destroyed the multiscale structures on the coated sample and decreased its surface roughness. The proposed method is the simplest method reported for controlling the mobility of water droplets on superhydrophobic coatings. The simple fabrication and operation of such surfaces facilitates their application in water transport and collection systems. The versatility of this simple spray-deposition technique was demonstrated through the fabrication of superhydrophobic coatings on various substrates. The photographs of water droplets on spray-coated substrates (including glass, metal, fabric, aluminum foil, paper, and plastic) are shown in Figure 6. The uncoated regions of the substrates were hydrophilic and could even be completely wetted, in which case the water droplets disappeared. However, the water droplets in the spray-coated regions were spherical and readily rolled off, regardless of the substrate.

## 4. Conclusions

In this paper, we describe a simple approach for the preparation of mechanically durable superhydrophobic coatings. Modified siloxane resins and silica NPs were sprayed onto the substrates. The dual-curing siloxane resins were synthesized via base-catalyzed hydrolytic condensation and cured with APTES. The use of optimal conditions for hydrophobic binder produced a coating with a surface energy of ~19.84 mJ/m² that could withstand scratching with a 9H pencil after curing. The effects of NP aggregation, NP concentration, and binder content on the hydrophobicity of spray-coated surfaces were evaluated. The superhydrophobic coatings were adequately robust and durable against severe environmental conditions such as mechanical force, high temperatures, and exposure to various organic solvents. The coatings were applied to various substrates over large areas in a water-collection system model.

## Figures and Tables

**Figure 1 polymers-12-01420-f001:**
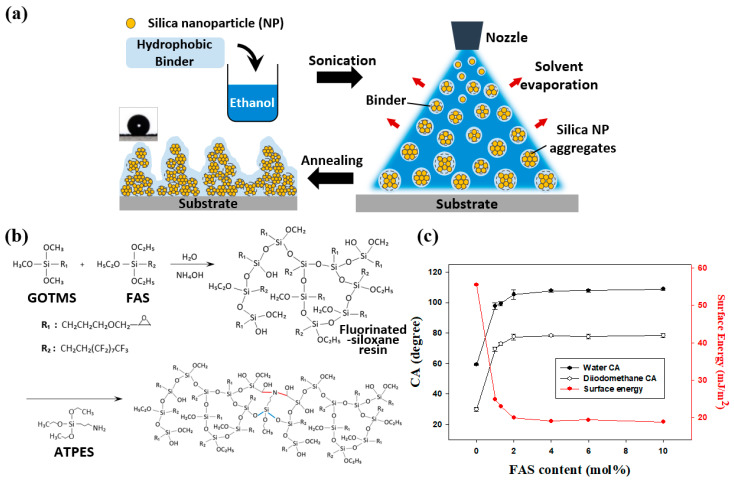
(**a**) Schematic of superhydrophobic surface fabrication with multiscale roughness by spray deposition. (**b**) Synthesis scheme and curing mechanism of a fluorinated siloxane resin with low surface energy. (**c**) Surface energies of spin-coated siloxane resins with FAS contents of 0, 1.0, 1.3, 2.0, 4.0, 6.0, and 10.0 mol %.

**Figure 2 polymers-12-01420-f002:**
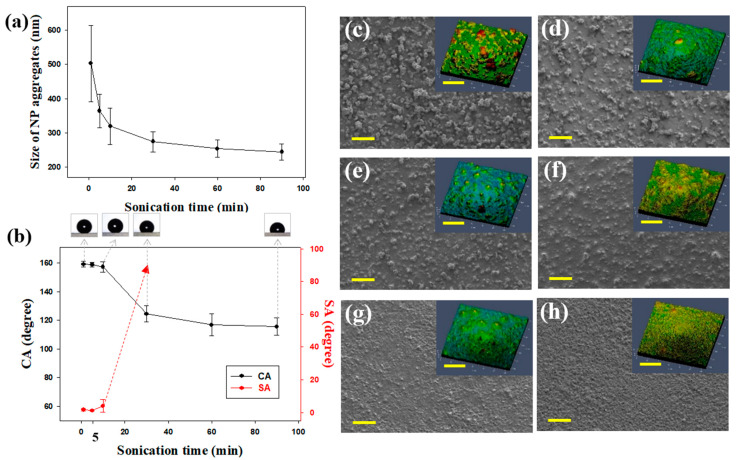
(**a**) Average sizes of silica NP aggregates measured by DLS after diluting the coating mixtures with ethanol. Measurements were performed after sonication for 1, 5, 10, 30, 60, and 90 min. (**b**) CAs and SAs of spray-coated samples with respect to sonication time. The red dotted line with an arrow indicates that SA measurement was not possible because a water droplet did not roll off at any angle. Static water droplets are shown above the corresponding sonication times. SEM and confocal laser scanning microscopy images of spray-deposited NPs on Si wafers after sonication for (**c**) 1, (**d**) 5, (**e**) 10, (**f**) 30, (**g**) 60, and (**h**) 90 min. Scale bar = 100 μm.

**Figure 3 polymers-12-01420-f003:**
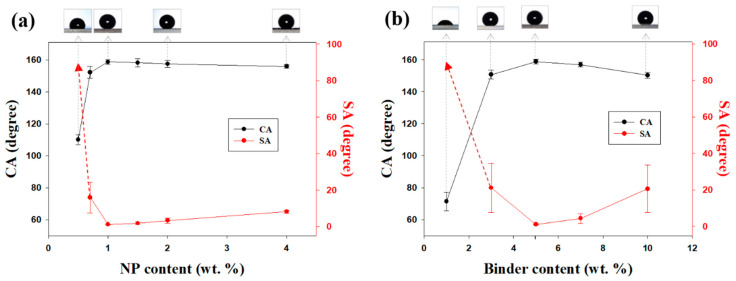
CAs and SAs of spray-coated samples with (**a**) NP contents of 0.5, 0.7, 1.0, 1.5, 2.0, and 4.0 wt. % with 5 wt. % of binder, and (**b**) binder contents of 1, 3, 5, 7, and 10 wt. % with 1 wt. % of NP content.

**Figure 4 polymers-12-01420-f004:**
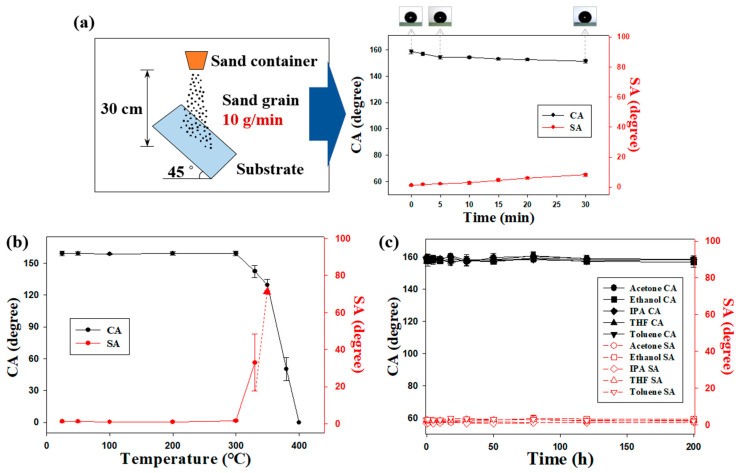
(**a**) Schematic of the sand abrasion test used to evaluate the robustness of coatings. The results of abrasion tests are shown in the right panel. (**b**) Thermal stabilities of superhydrophobic surfaces. (**c**) CAs and SAs of spray-coated surfaces after solvent resistance tests with acetone, ethanol, IPA, THF, and toluene.

**Figure 5 polymers-12-01420-f005:**
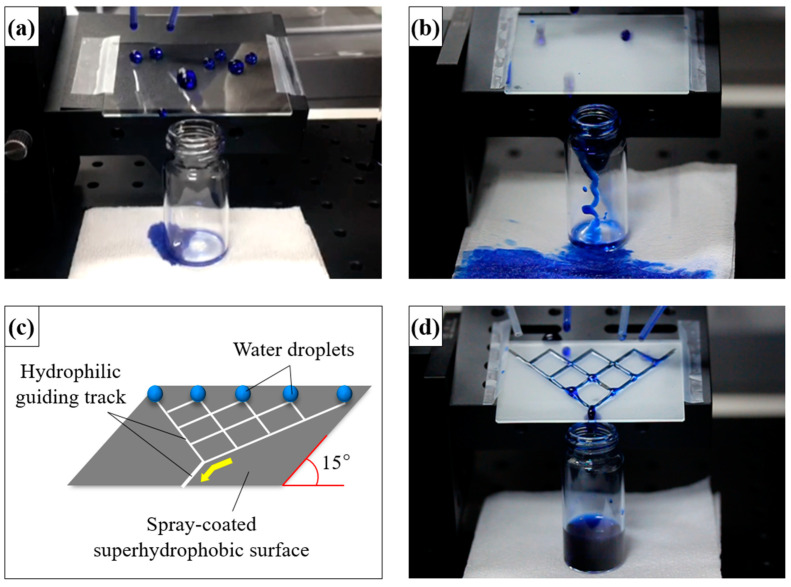
Transportation and collection of water droplets dyed with methylene blue on coated surfaces tilted at 15°. (**a**) Hydrophobic surface with no silica NPs; (**b**) superhydrophobic surface with NPs; (**c**) schematic of a superhydrophobic surface with hydrophilic patterns; and (**d**) patterned superhydrophobic surface.

**Figure 6 polymers-12-01420-f006:**
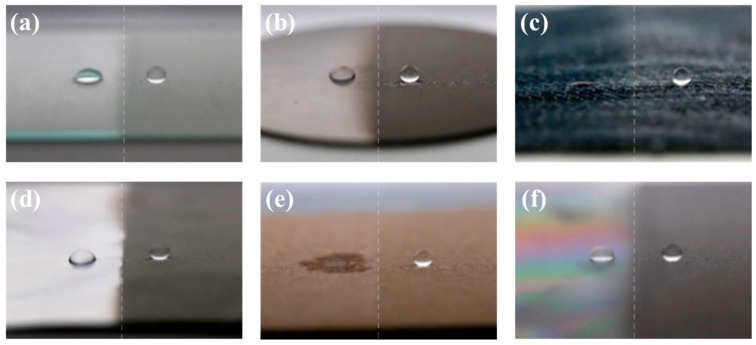
Photographs of water droplets on (**a**) glass, (**b**) metal, (**c**) fabric, (**d**) aluminum foil, (**e**) paper, (**e**) uncoated plastic surface, and (**f**) coated plastic surface.

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
