# Peer review of "Fabrication of Robust Superhydrophobic Surfaces with Dual-Curing Siloxane Resin and Controlled Dispersion of Nanoparticles"

_polymers, 2020, doi:10.3390/polym12061420_

Round 1

Reviewer 1 Report

Appreciate the authors for contributing an academic paper in the field of superhydrophobic surfaces.

A few minor comments and Suggestions as follow:

(1)What is the implication of binder content in Fig.3.?

(2)Please provide the data that the contact Angle of the water and the solvent mentioned in the article changes with time when the liquid drops on the surface of the coating stay for 30min, so as to further prove that this coating has a stable non-wettability.

Author Response

Point-by-point responses to the comments from the Reviewers

We appreciate these important comments. We have revised our manuscript based on these comments thank the Editors and Reviewers for their very positive and constructive comments on our manuscript. We now believe that the manuscript has significantly been improved. The revised sentences have been marked as red color. Please find detailed point-by-point responses to the comments by the Reviewers.

Reviewer #1:

Major Points

Point #1. “What is the implication of binder content in Fig.3.?”

Response: Thank you for this important comment. The binder content is the one of the important factors that affects not only CA and SA but also the pencil hardness of the superhydrophobic surface. The binder content in Fig.3b means the concentration of binder in sprayed solution with 1 wt% of NPs. The detailed explanation is written in paragraph of 3.2. wetting behavior. (Line 238-243)

 “Another variable affecting CA and SA was the amount of hydrophobic binder, which acted as an adhesive. The sonication time and silica NP content were held constant at 5 min and 1 wt%, respectively. The surface coating prepared without the binder was completely wetted with water droplets because the large number of hydroxyl groups on the surface of silica NPs made them hydrophilic. The CAs and SAs of spray-coated samples with binder concentrations of 1, 3, 5, 7, and 10 wt% are shown in Figure 3b”.

To make it clearly, we noted it accurately at the end of sentence as “Figure 3b”

Point #2. “Please provide the data that the contact Angle of the water and the solvent mentioned in the article changes with time when the liquid drops on the surface of the coating stay for 30min, so as to further prove that this coating has a stable non-wettability.”

Response: In this study, to demonstrate the durability and stability of the superhydrophobic surface, we conducted tests below.

Firstly, we measured CAs after a sand abrasion tests (Figure 4a, Line 257-263),

“To investigate the resistance of superhydrophobic coatings to mechanical force, sand abrasion and pencil hardness tests were performed. Figure 4a shows the schematic illustration of a sand abrasion test. Sand was released and allowed to fall freely from a height of 30 cm onto substrates coated with samples prepared under optimal conditions (section 3.2). The CA of water exceeded 150°, and SA fell below 10° after 30 min. However, the hydrophobicity of samples decreased only marginally. Sand abrasion results indicated that the fabricated superhydrophobic coatings were highly durable.”

Secondly, a submerging in different organic solvents (Figure 4c, Line 274-279)

“The resistance of the superhydrophobic coatings to organic solvents was assessed by exposing the samples to various environmental conditions. First, the spray-coated samples were submerged in common organic solvents (acetone, ethanol, isopropyl alcohol (IPA), THF, and toluene) for 200 hours. The samples were completely dried in an oven at 60 °C before measuring CAs and SAs on the coatings. Interestingly, the superhydrophobic surfaces retained their non-wetting properties after immersion in all aforementioned solvents (Figure 4c).”

However, we agree with the reviewer that we need further proving of a stable non-wettability. To clarify it, we conducted additional experiment. While staying the water droplet on the superhydrophobic surface for 30 min, the change in the CA was measured, and the result was added to “Figure S7”. Although, the water droplet was shrunk due to evaporation, the water droplet was not immerged into the superhydrophobic surface keeping the contact angle over 150 .

We added an additional information in that 3.3 Mechanical durability (Line 281-282), and “Figure S9” as shown below.

Revision 1.

Furthermore, to prove the stable non-wettability, CAs changes were measured while staying water droplet on the superhydrophobic surface for 30 minutes (Figure S7).” (Line 281-282)

Revision 2.

(figure -> Please see the attached files)

Figure S7. Photographs of CAs changes while staying water droplets on the superhydrophobic surface for (a) 0, (b) 5, (c) 10, (d) 15, and (e) 20, and (f) 30 min.

Reviewer 2 Report

The manuscript is nicely, concisely and adequately written. Apart from small spelling mistakes, which will be fixed in the proofing stage, I would recommend that the figures should be larger, or at least the text. Otherwise, I cannot find any fault within the paper. It was a pleasure to read such a nicely written paper!

Author Response

Point-by-point responses to the comments from the Reviewers

We thank the Editors and Reviewers for their very positive and constructive comments on our manuscript. We have revised our manuscript based on these comments thank the Editors and Reviewers for their very positive and constructive comments on our manuscript. We now believe that the manuscript has significantly been improved.

Response: Thank you for the comment. We found several mistakes as you noticed, and revise it as below.

Revision 1.

“The surface energies are plotted in Figure 1c as a function of FAS content and are summarized in Table S2.” (Line 166-167 in 3.1. Dual-curing siloxane resins with low surface energies)

was fixed as

“The surface energies are plotted in Figure 1c as a function of FAS content and are summarized in Table S1.” (Line 166-167 in 3.1. Dual-curing siloxane resins with low surface energies)

Revision 2.

“XPS analysis revealed a steady decrease in the F/O ratios on the coated substrate from 0.491 to 0.030 when the particle concentration increased from 0.0 mol% to 10.0 mol% (Figure S5).” (Line 237-238 in 3.2. Wetting behavior)

was fixed as

“XPS analysis revealed a steady decrease in the F/O ratios on the coated substrate from 0.491 to 0.030 when the particle concentration increased from 0.5 wt% to 4.0 wt% (Figure S5).” (Line 237-238 in 3.2. Wetting behavior)

Revision 3.

At last, we re-write the word in figures with larger size.

Reviewer 3 Report

I think that the subject is relevant, however the strategy is presented in a confused form and it is difficult to individuate the most important factors of control. Make a more simple and essential description.

On the point of view of the content I doubt about the possibility of controlling NPs dispersion by sonication. I would need to see a NPs average distribution for each sonication time.

If possible I would control the dispersion by functionalizing the silica Nps

Author Response

Point-by-point responses to the comments from the Reviewers

We thank the Editors and Reviewers for their very constructive comments on our manuscript. We greatly appreciate these important comments. We have revised our manuscript based on these comments and we now believe that the manuscript has significantly been improved. The revised sentences have been marked as red color. Please find detailed point-by-point responses to the comments by the Reviewers.

Reviewer #3:

Major Points

Point #1. “I think that the subject is relevant, however the strategy is presented in a confused form and it is difficult to individuate the most important factors of control. Make a more simple and essential description.”

Response: Thank you for the comment. In this study we present a simple method for fabrication of superhydrophobic surfaces by spraying a mixture of hydrophobic siloxane binder and silica nanoparticles. In response to your comments that it is difficult to identify most important factors of the studies due to unclear explanations, we have revised the following (1. Introduction, Line 69-74):

“Herein we describe a simpler and more efficient method for the fabrication of mechanically durable superhydrophobic surfaces. We prepared a robust siloxane resin as a spray-coating binder and controlled surface morphology with NPs. This method does not require a hydrophobic modification for NPs dispersion and a post-treatment process to lower the surface energy of the coated film. The dual-curing resin was synthesized via the hydrolysis and condensation of silanes and cured with amino silanes to increase durability. The surface morphology of spray-coated samples was regulated by controlling the dispersion of NPs in the coating solution by sonication time and the concentrations of coating materials. The fabricated superhydrophobic surfaces were exceptionally water repellent and highly robust. We performed pencil hardness and sand abrasion tests to evaluate the mechanical properties of the coatings. The superhydrophobicity of the coatings was maintained even at 300 °C, and they were highly resistant to various organic solvents.”

Point #2. “On the point of view of the content I doubt about the possibility of controlling NPs dispersion by sonication. I would need to see a NPs average distribution for each sonication time.”

Response: Thank you for the comment. In this study, NP’s DLS data in the solution and SEM and confocal laser scanning images after spray-coating were shown to determine the average distribution for each sonication time (1, 5, 10, 30, 60, and 90 min).

In the figure 2a, the average sizes of NPs in the solution by DLS were decreased along the sonication time. The average size of the silica NPs was decreased from ~500 nm to ~240nm. Furthermore, the figure 2c-h show the SEM and confocal laser scanning microscopy images of spray-deposited NPs on Si wafers after sonication. The particles have different aggregated size, but it has a regular dispersity on the substrate. To clarify the dispersion stability of the NPs in the solution, we conduct additional experiments (Figure S3 and Figure SA1). The aggregated size of the dispersed nanoparticles is almost the same as in Figure 2a, and it was confirmed that the dispersion is well maintained even if left for more than 2 hours. Then, we revised first paragraph in wetting behavior section as shown below based on your important comments. (Line 185-187)

Revision 1.

“The aggregates decreased in size with an increase in the sonication time (Figure 2a, and S3). When the sonication time was increased from 1 to 90 min, the average size of the silica NP aggregates decreased from ~500 to ~240 nm.” (Line 185-187)

Revision 2.

(figure -> Please see the attached files)

Figure S3. SEM images of NPs after sonication of solution for (a) 1, (b) 30, and (c) 90 min. Scale bar=1μm.

Revision 3.

(figure -> Please see the attached files)

Figure SA1. Photographs after leaving NPs (1 wt%) dispersed ethanol solution for (a) 0, (b) 5, (c) 10, (d) 20, (e) 60, and (f) 120 min after 1, 30, and 90 min of sonication treatment.

Point #3. “If possible I would control the dispersion by functionalizing the silica Nps.”

Response: We agree with the reviewer’s opinions. As reviewer’s comment, the dispersion of the NPs becomes different as functionalizing. In this work, the hydroxyl groups of silica NPs caused the homogeneous dispersion of NPs in ethanol, and a strong bonding with siloxane resins was induced through hydrolytic condensation reaction. Our method does not require a hydrophobic modification for NPs dispersion and a post-treatment process to lower the surface energy of the coated film. D.H.Lee and coworkers were shown a superhydrophobic surface with an aggregated silica particle modified by an octadecyltrichlorosilane (ODTS) treatment for 24 h more to give a hydrophobicity. (ref. 24, J. Mater. Chem. A 2014, 2(40), 17165-17173). Compared to this study, our spray coating method imparted hydrophobic properties to the surface without and additional functionalization process enhancing robustness. Furthermore, as mentioned in Introduction (Line 49-51), we could control the aggregation of NPs in a single solution.

“Particle aggregation can be controlled by selecting an appropriate solvent such as ethanol, propanol, or butanol [24]. However, few studies have addressed how NP dispersion affects the wetting properties of surfaces related to NP aggregation.” (Line 49-51, Introduction)

Furthermore, we conducted additional experiments to control the dispersion by functionalizing the NPs. To determine the effects of functionalization on the dispersion, silica NPs were functionalized with FAS, we took SEM images as you can see below as Figure SA2. After 30 min of sonication, silica NPs without any treatment were dispersed homogenously but FAS modified silica NPs were aggregated due to its hydrophobicity.

(figure -> Please see the attached files)

Figure SA2. SEM images of (a) NPs without any treatment, and (b) FAS modified NPs after sonication for 30 min. Scale bar=1μm.

Experiment for Figure SA2: (Heptadecafluoro-1,1,2,2-tetrahydrodecyl)triethoxysilane (FAS) was obtained from Gelest (Morrisville, PA, USA). 15–30-nm (ø) fumed silica NPs were purchased from Sigma-Aldrich (St. Louis, MO, USA). Absolute ethanol was purchased from Duksan Inc. (Ansan, South Korea). 10ml of ethanol with 1wt% of NPs were prepared, and 10  of FAS solutions were added. Solution was stirred at 200rpm for 6 hours prepared and stirred for 5 min.

Round 2

Reviewer 1 Report

appreciate the authors for supplementing the experiment data.

In my opinion, it’s clearer to characterize the NPs content as the content of NPs in the coating, NPs/ (NPs+ binder), and the binder content is also indicated out.

Author Response

Thank you for this kind and important comment. We try to revise it as below, the labels of the X axis were changed from NPs contents to NPs/ (NPs+ binder) as your suggestion (Figure A1), or NP (wt% with binder %wt%) (Figure A2).

(Figure A1)<- Please see the attached file.

Figure A1. CAs and SAs of spray-coated samples with (a) NP contents of 0.5, 0.7, 1.0, 1.5, 2.0, and 4.0 wt% and the (b) binder content of 1, 3, 5, 7, and 10 wt%.

(Figure A2)<- Please see the attached file.

Figure A2. CAs and SAs of spray-coated samples with (a) NP contents of 0.5, 0.7, 1.0, 1.5, 2.0,...

Unfortunately, we could not make figures clear and neat as changing labels. Therefore, we would like to suggest describing characterization of the NPs contents as a caption (Line 254~255). To simplify the visualization of the graph, we revised your mindful comment as below. I believe that, because of your revision, our study becomes more accurate and developed sincerely.

(Figure 3)<- Please see the attached file.

Figure 3. CAs and SAs of spray-coated samples with (a) NP contents of 0.5, 0.7, 1.0, 1.5, 2.0, and 4.0 wt% with 5 wt% of binder and the (b) binder content of 1, 3, 5, 7, and 10 wt% with 1wt% of NP.

(Line 254~255)

Reviewer 3 Report

Efforts has been made to organize the experimental description and simplify it. The replay of authors are satisfactory

Author Response

We really appreciate that your comments that could made our study more accurate, and improve description. Thank you.